# From Pathogens to Cancer: Are Cancer Cells Evolved Mitochondrial Super Cells?

**DOI:** 10.3390/diagnostics13040813

**Published:** 2023-02-20

**Authors:** Mario G. Balzanelli, Pietro Distratis, Rita Lazzaro, Van Hung Pham, Raffaele Del Prete, Adriana Mosca, Francesco Inchingolo, Sergey K. Aityan, Luigi Santacroce, Kieu C. D. Nguyen, Ciro Gargiulo Isacco

**Affiliations:** 1SET-118, Department of Pre-Hospital and Emergency, SG Giuseppe Moscati Hospital, 74120 Taranto, Italy; 2International Research Institute of Genetics and Immunology, Ho Chi Minh City 70000, Vietnam; 3Department of Interdisciplinary Medicine, Microbiology and Virology Unit, School of Medicine, University of Bari “Aldo Moro”, 70124 Bari, Italy; 4Department of Interdisciplinary Medicine, School of Medicine, University of Bari “Aldo Moro”, 70124 Bari, Italy; 5Department of Multidisciplinary Research Center, Lincoln University, Oakland, CA 94612, USA

**Keywords:** cancer, mitochondria, DNA, RNA, reactive oxygen species (ROS), superoxide anion (O2−, hydroxyl radical OH), electrons

## Abstract

Life is based on a highly specific combination of atoms, metabolism, and genetics which eventually reflects the chemistry of the Universe which is composed of hydrogen, oxygen, nitrogen, sulfur, phosphorus, and carbon. The interaction of atomic, metabolic, and genetic cycles results in the organization and de-organization of chemical information of that which we consider as living entities, including cancer cells. In order to approach the problem of the origin of cancer it is therefore reasonable to start from the assumption that the sub-molecular level, the atomic structure, should be the considered starting point on which metabolism, genetics, and external insults eventually emanate. Second, it is crucial to characterize which of the entities and parts composing human cells may live a separate life; certainly, this theoretical standpoint would consider mitochondria, an organelle of “bacteria” origin embedded in conditions favorable for the onset of both. This organelle has not only been tolerated by immunity but has also been placed as a central regulator of cell defense. Virus, bacteria, and mitochondria are also similar in the light of genetic and metabolic elements; they share not only equivalent DNA and RNA features but also many basic biological activities. Thus, it is important to finalize that once the cellular integrity has been constantly broken down, the mitochondria like any other virus or bacteria return to their original autonomy to simply survive. The Warburg’s law that states the ability of cancers to ferment glucose in the presence of oxygen, indicates mitochondria respiration abnormalities may be the underlying cause of this transformation towards super cancer cells. Though genetic events play a key part in altering biochemical metabolism, inducing aerobic glycolysis, this is not enough to impair mitochondrial function since mitochondrial biogenesis and quality control are constantly upregulated in cancers. While some cancers have mutations in the nuclear-encoded mitochondrial tricarboxylic acid (TCA) cycle, enzymes that produce oncogenic metabolites, there is also a bio-physic pathway for pathogenic mitochondrial genome mutations. The atomic level of all biological activities can be considered the very beginning, marked by the electron abnormal behavior that consequently affects DNA of both cells and mitochondria. Whilst the cell’s nucleus DNA after a certain number of errors and defection tends to gradually switch off, the mitochondria DNA starts adopting several escape strategies, switching-on a few important genes that belong back at their original roots as independent beings. The ability to adopt this survival trick, by becoming completely immune to current life-threatening events, is probably the beginning of a differentiation process towards a “super-power cell”, the cancer cells that remind many pathogens, including virus, bacteria, and fungi. Thus, here, we present a hypothesis regarding those changes that first begin at the mitochondria atomic level to steadily involve molecular, tissue and organ levels in response to the virus or bacteria constant insults that drive a mitochondria itself to become an “immortal cancer cell”. Improved insights into this interplay between these pathogens and mitochondria progression may disclose newly epistemological paradigms as well as innovative procedures in targeting cancer cell progressive invasion.

## 1. The Sub Molecular Structure of the Mitochondria and Cell Nucleus

The present knowledge and research efforts about cancer are almost completely centered on molecular and genetics investigation. The current paper’s aim is otherwise focused on the sub-molecular dynamics that support the sub molecular structure of the mitochondria and cell nucleus. This would open a fascinating scientific investigative research in which physics eventually would play a role in a better understanding of oncogenesis by highlighting the steady influence of pathogens, genetics, and external contributive factors such as diet and environment on cells and the mitochondria atomic structure. In this view oncogenesis would be seen as the amount of insults received and the mitochondria’s responses in trying to adapt and survive a destabilized atomic structure composed of elementary particles of protons, electrons, and the fermions [1,2].

The spin has a fundamental role in the nature of matter’s structure as an intrinsic property of particles connecting each other in the presence of magnetic fields. Electrons, while incessantly moving around nuclei or between atoms/molecule rotate about their axis generating a magnetic field (in physics any movement charge produces a magnetic field) called angular momentum or spin. The electron spin value is ±½ depending on the direction of its rotation. The Pauli exclusion principle asserts the impossibility of existence of two identical electrons in the same state, electrons are paired in the atomic structure only if they have an opposite spin value. Without the Pauli exclusion principle, chemistry would not have the Periodic Table [3,4].

The atomic stability of matter’s structure which indicates the precise order of electrons and nuclei in atoms and molecules, is strictly based on spin as a crucial feature which has physical significance in chemistry. The spin, although being purely quantum-physical, is a solid foundation in the real-world, in this case the micro-scale system has tangible consequences in large-scale systems such as in living tissues and organs. In chemistry those molecules containing a single, unpaired electron are known as free radicals, which are strongly unstable elements and chemically highly reactive (Figure 1) [5,6].

The synthesis of the almost totality of proteins and complex molecules requires the oxidation of their precursor, via the use of molecular oxygen, vital in many biological pathways. The aerobic organisms’ capacity to harness the power of molecular oxygen as a terminal electron acceptor in their respiratory cycles played a crucial role in the life evolution process. Oxygen presents a diradical nature, oxygen-based radicals are referred to as radical reactive oxygen species (ROS) and superoxide anion (O_2_−, hydroxyl radical OH). Electrons are substantial parts of molecules showing opposite trajectories, spinning in different orbitals taking part in bond formation as well [5,7].

Mitochondria are a major source of endogenous ROS produced at cell level. The uncontrolled accumulation of free radicals can damage macromolecules as well as the microstructure such as nucleic acids, proteins, and lipids advancing slowly to general tissue and cell decay typical of chronic, degenerative diseases and cancer. Antioxidants play a crucial role in the body’s defense against free radicals. An adjunctive aspect in this picture is the role that a few enzymes such as superoxide dismutase (SOD), glutathione peroxidase (GPx), and catalase (CAT) may eventually play in the energy mechanism obtaining and storing available energy from oxidation processes, important in adenosine-triphosphate (ATP) intracellular high-energy-related compartments. In addition, redox reactions may affect multiple phases including the signaling between molecules bound to DNA with a potentially key effect on the cell cycle, up to the production of free energy from the dissociation of a water molecule [7,8].

Several outcomes have pointed out the connection between persistent inflammation, oxidative stress, and carcinogenesis; those studies revealed the presence and relationship between Epstein–Barr virus/human papilloma virus (EBV/HPV) coinfection, Chlamydia trachomatis, and enzymes such as GPx, SOD, and CAT level in several types of cancer. SOD, the most important antioxidant enzyme in aerobic cells, is in charge to eliminate superoxide radicals and catalyzes the dismutation of hydrogen peroxide and molecular oxygen. The GPx, the key of hydrogen peroxide-scavenging, converts this molecule to water, both work to maintain oxidants level within acceptable limits, protecting cell and mitochondria DNA damage [8].

Surely, one of the most difficult attempts was to clarify the effect of different spin modality on the bio-molecular system. Probably the most known in the field of oncogenesis is the series of influences on modulation and control of different tumor suppressor proteins and regulators such as the case of p53 protein and Nfr2, both important tumor suppressor factors on different promoters using DNA-mediated electron transport [9,10].

## 2. The Bacterial Origin of Mitochondria the Common Genetic Traits with Virus and Bacteria

Microorganisms constantly evolve finding always a new solution to survive and adapt to environment restrictions, these changes will be then quickly encoded into their genetic make-up. Microorganisms rapidly adapt faster than plants, animals, and humans, including by genome reduction and horizontal gene transfer.

Given the huge complexity of the changes involved, the evolutionary transition from the early stage of pre-mitochondrial alpha-proteobacterium into the mitochondrial ancestor of modern mitochondria was necessarily characterized by multiple incremental complex interphases and forms. Notably, of the 1000 proteins identified in mitochondria, 40% are of bacterial origin. To trace back mitochondria’s origin, scientists carried out experiments with Jakobida, a sort of alpha-proteobacterium, an early diverging eukaryotic clade bearing the most bacterial-like mito-genome [11]. The Jacobida internal system underwent a series of changes that were compatible for its use in eukaryotic cells. In fact, the Jakobida, according to the authors, still keeps a mitochondrial tRNAGly isoacceptor in a state of transition, something like in between a bacterial type with U73 and a eukaryotic type A73 containing eukaryotic-type mito-tRNAGly [11]. Scientists found two important switchers in the translational system, one in the discriminator recognition code of a chiral proof-reader DTD (d-aminoacyl-transfer RNA (tRNA) deacylase) and the other in mitochondrial tRNAGly that functions as a catalyzer enhancing the compatibility between different factors crucial for switching on the survival mechanism. A change was found in the discriminator element for mitochondrial transfer RNA (tRNA) that shuttles glycine, a key amino acid for making a protein. The mito-tRNA-Gly discriminator element is unique in its ability to switch from pyrimidine to purine during the bacteria-to-mitochondria transition, a move characterized from a change of U73 nucleotide (uracil) to A73 (adenine), in order to be compatible with eukaryotic DTD. Nevertheless, it still remains unclear how the DTD prevent smisediting Gly-tRNAgly [11,12,13,14,15,16].

Further modifications during the transition were at the time of organellogenesis in this rudimentary system. According to the outcomes, this was the crucial moment in which many of the products essential to organelle activities of the endosymbiont-encoded genes were steadily moving into the eukaryotic nucleus. Similarities are still traceable in modern mitochondria, the SAM, TIM23-PAM, OXA, and MPP complexes, members of the major import protein pathway were confirmed to have alpha-proteobacterial homologs with additional subunits acquired during the millennia of evolution. Intriguingly, scientists showed that SAM and OXA complexes work in similar ways as their ancestral alpha-proteobacterial counter-parts, whilst both TOM and TIM23-PAM complexes were shown to use functions from the time of organellogenesis [17,18,19].

A further phase was to ensure the insertion of the mitochondrial ancestor compartments into clone cells and progenies during cell division, controlling the distribution throughout each cell. At that time pre-mitochondrial alpha-proteobacterium cloned thanks to the contractile Z ring, composed of polymerized FtsZ protein, controlled by a mutually antagonistic system of Min proteins, and, based on the protein anchoring ring to the cell membrane. Of these components, the mitochondrial ancestor retained at least the FtsZ protein (which underwent duplication prior to LECA), and the three Min proteins, whilst acquiring an external dynamic ring that helped fastening at the mitochondrial mid-point [20,21,22].

## 3. How Pathogens May Affect the Mitochondria Stability

Some bacteria and viruses evolved the ability to remain in specific cells for long periods of time. These infections include the following: latent, chronic, and slow virus infections. The type of persistent infection usually influences the extent of cellular changes [23,24,25].

This view can have a profound impact on how we view viruses and bacteria such as Chlamydia trachomatis, human papilloma virus (HPV), Epstein bar virus (EBV), herpes virus (HV), HIV, helicobacter pylori (Hp) which interact with host cells and mitochondria and how the latter respond in trying to defend against these pathogens.

Chlamydia trachomatis, Hp, HPV, EBV, HV, and HIV are very common pathogens that can often remain symptomless, the majority of times resolved within 2 years without therapy. However, in some cases, all of them can cause precancerous lesions, which, if not removed, increase the risk of cancer in the affected area in stomach, in genitalia, in the mouth and in the intestines. Latent infections are characterized by restricted expression of the episomal or integrated virus genome. The pathogen’s genomic product(s) are associated with few, if any, changes in the latently infected cell. Chronic infection: The cellular effects of chronic infection are usually the same as those of acute cytocidal infections, except that production of progeny may be slower, intermittent, or limited to a few cells. The long-term cellular changes may result in severe diseases, immune suppression, or may trigger immune responses to damaged, or undamaged cells or tissues. Slow infection: This type of virus/bacteria-cell interaction is characterized by a prolonged incubation period, without significant morphological and physiological changes of infected cells. A slow progression of cellular injury may take years and is followed by extensive cellular injury and disease [23,24,25]. This shows that there may be a different etiology, or different mechanisms through which these pathogens can lead to pre-cervical cancer in individuals. It may be that the vaginal microbiome plays a stronger (perhaps protective) role in White women than Black women At a very basic level, the mitochondria similar to virus and bacteria needs to re-program metabolism back to a biosynthetic mode, which could be a very old strategy indeed, while manipulating the immune system. If the original mechanism of the Krebs’s cycle was biosynthetic, using the energy in a proton gradient to build molecules, a later evolutionary step was based on the reverse mode by taking energy out of molecules to generate life. One of the key mechanisms for doing this relies on electrons flowing to a highly electro-negative compound or element, such as oxygen, the present-day metabolic pathway between inflammation, biosynthesis ,and hypoxia, reflecting glycolysis and mitochondria functionality. This shows that there may be a different etiology, or different mechanisms through which these pathogens eventually lead to pre-cervical cancer to cancer though the changes and mutations are always seen as a response to constant inflammatory insults (Figure 2) [23,24,25].

Of note, the mitochondria are the only non-nuclear organelles with their own genome, encoding 13 polypeptides of the oxidative phosphorylation (OXPHOS) subunits and respiratory chain and two ribosomal RNAs with 22 transfer RNAs necessary for translation of peptides and polypeptides into protein compounds. However, the oncogenesis seen as a response to bacteria and virus steady insults relies on its unique feature as a multi-task independent unit, regulating ion channels (calcium, magnesium, sodium, and potassium) and reactive oxygen species (ROS) homeostasis; mitochondria are the major source of ROS production units within the cells, a capacity that dates back from the time they assembled with an Archean protocell to become a eukaryote [20,21,22,23,24,25].

Free radicals are completely unstable, short-lived, and highly reactive molecular fragments due to the odd number of electrons, characterized by complete autonomy and capable of carrying one or more unpaired electrons in an outer atomic or molecular orbital (hence either “free” or “radical”) [25]. Free radical unpaired electrons are denoted by a dot on the atom or group in which it is located, for example •H (hydrogen atom), •OH (hydroxyl), •CH3 (methyl), and •CH2CH3 (ethyl). In oxygen radicals, the unpaired electrons are located predominantly on an oxygen atom, for instance the superoxide (O2•−), the hydroxyl (•OH), and peroxyl (ROO•). A free radical’s important feature is its highly reactive nature due to the odd number of electrons and responsible for chain reactions as it always tries to attempt to bond with other molecules, atoms ,or even individual electrons to create a stable compound by using succinate [26].

A typical mitochondrial anti-pathogen activity takes place with the use of fumarate and itaconate which are either anti-viral or immunomodulators and are capable of controlling the nuclear factor erythroid 2-related factor 2 (Nrf2), a key regulator of oxidative stress, and indirectly the inhibitor of succinate dehydrogenase (SDH). Of note, the SDH works as catalyzer of succinate oxidation to fumarate in the Krebs cycle, releasing electrons to the ubiquinone (UQ) pool of the respiratory chain setting, an electrochemical gradient across the mitochondrial inner membrane allowing ATP synthesis [26,27,28].

However, this mechanism may display multiple aspects either as a self-defense anti-pathogen tool or it may function against itself if not properly controlled. This might suggest that looking at succinate it could be important to assess the rate of inflammation and the disturbances that eventually affect the sub-molecular stability. Interestingly, activators of Nrf2, such as dimethyl fumarate have indicated that some viruses (SARS-CoV2, HPV, EBV, HV, and HIV) or bacteria (Chlamydia, Mycobacterium tuberculosis, M. leprae, Listeria monocytogene, Brucella spp, and Hp) may suppress or interfere with the Nrf2 pathway, suggesting feasible and alternative therapeutic solutions, as this pathway is not only anti-inflammatory but appears to have a distinct anti-oncogenic ability via modulating ROS. On comparing patient and healthy individual metabolomic studies they revealed a level of higher succinate and lactic acid, but lower citric acid levels on affected subjects which was indicative of Krebs cycle modulation and mitochondrial dysfunction (Figure 3) [28,29].

The oncogenic role of the overexpressed Nrf2 is thought to be related to its facilitation function for mitochondrial activity and metabolic reprogramming to meet the increased energy demand of rapidly growing cancer cells. It is widely accepted, that oxygen levels and metabolism are very tightly integrated. Interestingly, many viruses, bacteria, and fungi are highly sensitive to oxygen tension, but the outcome is very context dependent, and the response varies with the species and the host cell. Hypoxia enhances virus and mycobacterium replication worsening inflammation milieu via the ORF3a mechanism, directly affecting mitochondria which slowly tend to switch towards a glucose type of energy source, more and more pronounced by the hypoxia inducible factor (HIF)-dependent mechanism. This could be seen in both macrophages and monocytes and enhances glycolysis inhibiting T cell responses [28,29,30,31,32].

Within the Krebs cycle, succinate is involved in signaling, tumorigenesis, inflammation, redox, and epigenetics but it can also suppress viral and bacterial replication in the nucleus by succination and via tumor necrosis factor (TNF), driving reverse electron transport (RET) through complex-1 generating ROS. An interesting point on both bacteria and viruses is that whilst managing the immune system, virus and bacteria can replicate via the modulation exerted on the Krebs cycle by which they receive the metabolites they need, such as citric acid [28,29,30].

## 4. Molecular and Cellular Heterogeneity of Tumors

The major obstacle to cancer treatment and personalized medicine is the extreme heterogeneity of the cancer molecular core, represented by a multitude of cells, distinctive and diverse patterns, and microenvironments. Investigating and clarifying the intratumor heterogeneity may thus give a better understanding towards a new approach either in prevention or in predicting the risk of cancer progressive resistance [33].

Nevertheless, intra-tumor heterogeneity cannot be limited solely to those aspects neither to only genetic paradigms. Several outcomes confirmed that the effects of the same genetic variant are often different across populations, realigning the precision of medicinal efforts to acknowledge the challenges and to optimize the opportunities this paradigm shift represents. Eventually cancer cell lines with a high degree of genetic homogeneity responded to drugs also based on intercellular epigenetic heterogeneity [33]. Epigenetic mechanisms involve numerous processes as well, including DNA methylation, post-translational modification of histones, and the effects of single nucleotide polymorphisms (SNPs) [33].

Intratumor heterogeneity may define specific traits and differences between cancers of the same origin in different patients. The therapy success is articulated as well and depends on achieving sustainable and continuous results, which is often a matter of having understood the complexity of the heterogeneity, the specific individual molecular signatures, and therefore the different biological behaviors [33].

The proposed models of inter-tumor heterogeneity are based on, (i) the genetic and/or epigenetic mutation model for which mutations/modifications and (ii) the cell-of-origin model, by which different cell populations in the lineage hierarchy are considered as the origin that primarily determines the phenotype of the tumor and different tumor subtypes.

However, these models are still not conclusive. Therefore, we proposed a third model which we believe includes the previous two making them more conclusive. The third model suggests cancer cells as super cells derived directly from mitochondria through years of adaptation and mutation in response to internal and external insults (internal and external epigenetic stimuli as proposed by the 1st and 2nd models). Eventually, it seems that mitochondria differentiation toward cancerous cells is marked by several molecular and genetic reprogramming steps and phases over a long period of time. By this, mitochondria with the support of either endogenous or exogenous mechanisms tend to organize a definitive structural displacement from the host cells to become completely autonomous. This achievement of structural autonomy is heritage encoded and therefore promoted by their own genetic make-up, something that mitochondria share with bacteria and viruses. The same as for bacteria and viruses, mitochondria went through indefinite numbers of evolutionary stages from free-living archeo-bacteria to become an integrated bio-system of the host cell. From this symbiosis both parts gained something, host cells evolved into modern eukaryote lineages, while mitochondria reached the capacity of performing higher and complex functions; eventually both evolved sharing integrated division mechanisms. Given the huge amount of time that viruses, bacteria, and mitochondria hosts have been coexisting, it is no surprise that the mitochondria have evolved as well and through a highly metabolic reprogramming process have reached the capacity of nullifying their threats to survive at the expense of the host cell. This would support a polyphyletic origin of the mitochondrial Krebs cycle, and raises the question of what forces drove the evolution of the mitochondrial proteome [11,32,33,34].

The scenario discloses what the mitochondria’s metabolic defense program indicates as “pathogens’ steady manipulation” as an effective strategy adopted by mitochondria to survive. We can consider that the Warburg shift benefits both the host and the pathogens whilst raising a mild hypoxia as a trick used by mitochondria through their evolution. In this perspective, there is an evolutionary rationale as to why mitochondria would modulate Krebs cycle intermediates to ensure their own replication, including lipid and protein metabolism, equally seen as a behavior to control pathogens aggressiveness. Nevertheless, this strategy could be truly ancient indeed, as theories on the earliest proto-metabolism that gave rise to life indicate that they can lead to the formation of basic amino acids, proteins, and lipids [32,33,34,35].

## 5. A Preclinical and Clinical Perspective

The fact that mitochondria are able to reprogram themselves in trying to contrast harmful pathogens and toxin loading leads to serious repercussion to the system, well explained by the presence of inflammation driven chronic phase responses (CPR) and altered metabolism which contribute to disorders such as insulin resistance, degenerative patterns, and tissue alterations. Some of the proteins released during the CPR can actually open up inflammasome activation, hinting that the host is trying to escape from prolonged excessive inflammation. Consequently, the tiny boundary between “directly pathogen induced” vs “host response to pathogens” not only becomes clearer as an interchangeable process but it confirms the autonomous pleotropic nature of mitochondria. The metabolic link is further strengthened as these pathogens not only worsen during existing metabolic disorders such as diabetes, arthritis, or neurodegenerative processes but can also induce them in people who never previously suffered from them [36,37,38].

A further distinctive trait of cancer cells is cell surface negative charge, which is metabolically regulated by glycolysis, an energy up-taking process depending on the level of glucose provided. The negative charge is substantial and based on the alternative sugar metabolism pathways, conditioned by the ionic microenvironment in which cancer cells are located. The negative charge is crucial in determining cancer cell rate of proliferation, state of aggregation, and adhesion in the different organs and tissues. According to the current scientific position, in this way viruses as well as different intracellular bacteria are able to take over the host metabolism and start to reprogram the host cell’s DNA of either nucleus or mitochondria to ease pathogen reproduction. Thus, DNA viruses and bacteria can cause alterations in several metabolic pathways, including aerobic glycolysis known as the Warburg effect, based on pentose phosphate pathway activation, and amino acid catabolism such as glutaminolysis, nucleotide biosynthesis, lipid metabolism, and amino acid biosynthesis. The interference with internal affected cells allows the necessary energy for the reproduction mechanism via modification of a variety of carbon source utilizations [39,40,41].

In some aspects, this could be considered as a consequent highly-specific metabolic reprogramming effect which involves both the mitochondria target structure affecting the sub-molecular atomic stability as a consequence of ROS augmentation and the electron shift disabling the vital energetic mechanism from the generation of ATP in favor of fuel autophagy. In highly aggressive tumors, the mitochondria, similar to bacteria and virus, are able to reprogram their energy pathways to meet the incredibly high demand of energy needed for a rapid cell division and migration. In this way mitochondria are able to regulate and settle the oncogenic system mode via the mitochondria-to-nucleus signaling mechanism taking over the command of newly formed cancer cells. The potential issues are highlighted by the mitochondrial-lysosomal theory of aging. Of note, as mitochondrial function decreases and thus the ATP production to drive the vATPase, the ability to remove damaged components, including sub-functional mitochondria by autophagy decreases and could lead to a vicious cycle. However, non-autophagocytosed mitochondria undergo further oxidative damage, resulting in decreasing energy production and increasing generation of reactive oxygen species. Consistent with these observations, different outcomes were able to demonstrate the mechanism adopted by breast cancer cells which interfere in the small GTPase Arf6 -based pathway inducing the localization of ILK to focal adhesions to block RhoT1-TRAK2 association, which controls mitochondrial retrograde trafficking [42,43]. Intriguingly, the inhibition of the RhoT1-TRAK1 machinery (mitochondrial trafficking proteins RHOT1 and RHOT2, the removal of which may be required for immobilization of mitochondria prior to mitophagy) impaired cell invasion, but not cell migration. Nevertheless, in cells that do not show invasiveness or are weakly invasive, the TRAK proteins are undetectable [41,42,43].

In addition, as confirmed by previous outcomes the disturbances in mitochondria genetic make-up also favor the oncogenesis process. For instance, it was confirmed that the presence of SNPs in those genes regulating the uncoupling protein 1, 2 and 3 (UCP1, 2 and 3) members of the super family of anion carrier proteins, located in the inner membrane of mitochondria, are linked with mitochondria-related metabolism. The outcomes revealed that any damage or interferences tends to disrupt cellular homeostasis contributing to cancer formation. Different studies otherwise confirmed that ectopic expression of ATP6, responsible for the final step of oxidative phosphorylation in the electron transport chain, carries mutation (T8993G or T9176C) induced tumor growth through inhibition of apoptosis. Another study showed how specific mutations in mitochondrial tRNA genes were distinctive traits of mitochondria deviances toward cancer progression. The study indicated that five tRNA mutations (mt-tRNAAla, mt-tRNAArg, mt-tRNALeu, mt-tRNASer, and mt-tRNAThr) rarely found in healthy specimens, were instead present as tRNA metabolism disturbances in lung cancer. Additionally, Meng et al. reported that tRNA mutations (mt-tRNAASP) caused a tertiary structure and led to impairment of mitochondrial protein synthesis in breast cancer. SNP presence on two noncoding genes (MT-RNR1 and MT-RNR2) encoded by mitochondrial DNA, mainly responsible for mitochondrial protein synthesis, were found to act as prognostic markers especially in hepatocellular carcinoma; the poor prognosis of HCC is largely due to high rates of tumor metastasis. Data from those studies confirmed that the expression levels of the glycolytic regulator hexokinase 2 (HK2) were higher in the MT-RNR1 709A group than in the MT-RNR1 709G group [43,44].

The OXPHOS pathway t in normal condition represents the main source of energy in eukaryotic cells. This process is performed by means of electron flow between four enzymes, of which three are proton pumps, in the inner mitochondrial membrane. The mtDNA-encoded OXPHOS genes are essential for cancer cell survival and growth, mutations in mitochondrial genes must therefore alter cellular bioenergetics and metabolism in ways important for neoplastic transformation.

Damaged, enlarged and functionally disabled mitochondria gradually displace normal cell nuclei replacing them, which cannot replicate indefinitely due to intrinsically limited cell volume and, here the differentiation process to cancerous cells would eventually commence [44,45].

## 6. Final Consideration when Bio-Medicine Meets Physics

Thus, in this current oncogenesis theoretical view there is a shift of paradigm, the mitochondria reverse their own status and go back to their original state as independent bacteria, following the rule of adaptation and survival. In general, biology explains this event as a reversible process in which a system moves from a state of initial equilibrium to a state of final equilibrium, through sequences of regular modifications which keep a constant thermal and mechanical balance with the microenvironment being extremely slow and almost imperceptible. However, it is generally accepted that in any biological system returning to an initial state via a material process, e.g., transient enhanced mitochondrial to proto-bacteria, the quantum state entropy should also increase without changing the entropic functionality. In response to primary incoherent interatomic status which inexorably follows, there will take place sequential dysfunction of the microstructures, and disequilibria in small volumes of tissues. Instability of molecules and molecular assemblies then induce (1) genetic misinformation transfer errors (presence of SNPs for example), and (2) immunogenic microbiological and pathogenic hyper-responses that may elevate the inflammatory microenvironment that destabilizes the quantum state normality [46,47,48].

Enhanced tumor growth within the first settled changed microenvironment, under the influence of an inhomogeneous sub-molecular magnetic field, involves the uncontrollable reaction of free radical reactions and subsequent redox signaling (ROS high). Thus, it seems that quantum mechanics plays an important part in clarifying the initial process that promotes mitochondria differentiation to cancer from a free radical mechanism in tumor redox-mediated pathways in which, either external or internal magnetic fields act on the entangled states of electron spins in building up free radical pairs within the mitochondria and host cells [1,2,3,4,5].

The particle-wave equation of Jacobson Resonance for instance, has yielded a testable theoretical model for the mechanism of magnetotherapy with the equation mc2 = Bvl coulomb which sets in dual resonance gravitational and electromagnetic potentials. A theory that proposed the unification of two physic’s law cornerstones, Einstein’s gravity and Maxwell’s electromagnetism. According to this theory, it is widely accepted that well-known genomic magnetic domains are oncogenes with associated structures like peptide hormone trophic factors [48,49].

In chemical and biological systems, Rudolph Clausius proposed the following equation:**ΔЅ ≥****∫ ^final dq/initial T^**(1)
where T is the absolute temperature at which the change in heat (dq) occurs and DS is the change in entropy. Any system becomes progressively disordered (with increased entropy) as its temperature rises. Equation (1) holds for processes in which the system remains in equilibrium throughout the change. These are known as reversible processes. For biological processes, Equation (1) reduces to the following:**ΔЅ ≥ q/T**(2)

There is an increase in network entropy in the oncogenesis microenvironment and, the gene expression differences between normal and cancer cells are characterized by an anticorrelated status with local network entropy changes. During this phase it seems that genetics errors and mutations are stronger, increasing cancer cell growth and development with a consequent increased energy use necessary for their mechanical processes [50,51].

Of note, in thermodynamics, entropy is strictly correlated to a system’s energy use that is necessary in mechanical processes. Atoms and molecules are able to either absorb or emit just a proper amount of energy. Energy is quantized; it occurs in fixed quantities, rather than being continuous. Atoms modify their energy state by shifting electrons and by modulating quanta of energy by either absorbing or emitting it [50,51].

In cancer, the system becomes apparently chaotic, however we assume that this phase reflects the transition of mitochondria back to their original state, in fact the heterogeneous models which use more energy (ΔE) from the normal existing systems are based on the need to continue and sustain the ongoing performance. Scientists have confirmed that this process, named a fissionary process, will affect normal structures, deactivating them to cope with an ever-increasing number of sub-molecular instabilities and collisions producing polyploidy. An increasing level of configurational energy produces an increasing level of disorder of the biostructures which is reflected by the huge amount of Ca^2+^ and Mg^2++^ accumulated within the cancer intracellular compartment. In a cancerous deconstructed system, the expression of photomechanical forces generated by distorted electric forces propagates beams in a highly linear substratum (Figure 3 and Figure 4) [51,52].

## 7. Current Advances

The key consideration in targeting mitochondrial mutation is that normal cells use mitochondrial ATP production for survival, therefore the therapeutic strategic moves may be quite limited. The only exception would be if we reach the capability of recognizing cancer cells that selectively reveal highly peculiar features, either atomic instability determined by electrons displacement, or particular uptake of the inhibitors of mitochondrial ATP production compared to normal cells, or specific mutation that could be selective by using antagonist bacteria or antibiotics. On the basis of recent evidence, we suggest that some antibiotics and particular bacteria strains may be considered as viable anticancer agents that target mutated mitochondria cancer cells characterized by high ATP production without provoking toxicity in normal tissues [53,54,55].

Previous studies collectively indicated that specific traits of cancer cells, such as invasion, proliferation, contamination and, resistance against therapies, including tolerance to ROS, are shared with virus and bacteria. In fact, currently the use of specific antibiotics in cancer treatment has shown a few appreciable results though the side-effects need to be taken into account.

These types of antibiotics induced apoptosis of cancer cells selecting specific targets such as the apoptotic gene B cell lymphoma-2 (Bcl-2), apoptotic pro-Bcl-2-associated x (Bax), caspase-3/8/9, and cancer suppressor gene P53. Interestingly, some of those antibiotics were shown to perform as regulatory agents of anti-epithelial-mesenchymal-transition (EMT) inhibiting metastasis. Ciprofloxacin for instance was revealed to have clear pro-apoptosis ability while salinomycin was characterized by the inhibition capacity for proliferation and EMT in the development of cancers [54].

The use of a particular bacteria strain that may work as an antagonist has shown promising results. Data from studies confirmed that patients with lung and kidney cancer showed low levels of the bacterium *Akkermansia muciniphila*. Interestingly, the fecal samples from non-responding melanoma patients receiving anti-PD-(L)1 therapy showed an imbalance in commensal bacteria composition which was linked with impaired activity of immune cells. Oral supplementation of *Bifidobacterium* alone was shown to improve tumor control to the same magnitude as anti-PD-(L)1 therapy, and achieved in combined therapy combination a marked reduction of cancer outgrowth due to an increase in dendritic cell activity together with an enhanced response of CD8 + T cell within the tumor microenvironment. Similarly, the presence of distinct Bacteroides was shown to have a strict relation with the anticancer effect of Cytotoxic T-Lymphocyte Antigen 4 (CTLA-4) [53,54,55].

In addition, we proposed something that may eventually help either diagnosis or treatment. During the last twenty years scientists increased their efforts in searching for a physical or mechanical difference that could help to center highly selective markers between cancerous and normal cells. They found important distinctive patterns, the surface coat surrounding cancer cells revealed what we know under the definition of “fractal dimensionality.” Fractals are frequent physical events that occur as a consequence of chaotic behavior and cancers have been always associated to a chaotic condition. Those effects were substantially observed in the mitochondria and at different cell levels as a new hypothesis for heavy non-radioactive isotope fractionation in living systems via neutron effect realization [1,56]. The results obtained after comparing different stages of cancer versus normal tissues from the same organ, such as pancreatic, breast, colon, and prostate confirmed the fractal evolution of cancer from stage I to stage III with an overall increase of 7% [57].

## 8. Conclusions

We propose that cancer cells are mutated mitochondria, the final result of a very long term metamorphosis that mitochondria as archeo-bacteria are able to go through to survive from a steady general cellular decay. This conceptualization brings many scientific and philosophical implications and would also explain the mitochondria/cancer cells absolute capacity to respond to all immunity attacks with a high level of adaptation to any adverse circumstances. From our perspective, a significant point was to determine and characterize whether this metamorphosis would take place in different levels and via multiple phases, at the atomic level, via metabolic changes, and thanks to genetic alterations. Whether victims or killers, mitochondria are to be found right in the center all human health issues and treatment possibilities. They still resemble their bacterial ancestors and still heavily maintain some independence from us. However, due to the deal they made with our archaea cells a long time ago, their life is entwined with ours. Nevertheless, we have tried to answer if these slowly but steady transformations toward cancer cells could be possibly categorized as a unique process that could eventually be seen as a survival effort. During the last two decades, researchers have gained new evidence and information and increased the understanding of the developmental origins of heterogeneity found among human cancers using biogenetic-physics models. These approaches will surely enhance the comprehension on which a mechanism among the multitude mechanisms that are part of the human body could better describe the effective transition towards cancerous cells as well as to other single-cell transcriptomic profiling in development and disease. These technologies will become valued complements to new employing strategy models created to support conventional research in genetics, immunology, and microbiology, contributing to a concerted effort to understand and to treat cancers as successfully as possible.

## Figures and Tables

**Figure 1 diagnostics-13-00813-f001:**
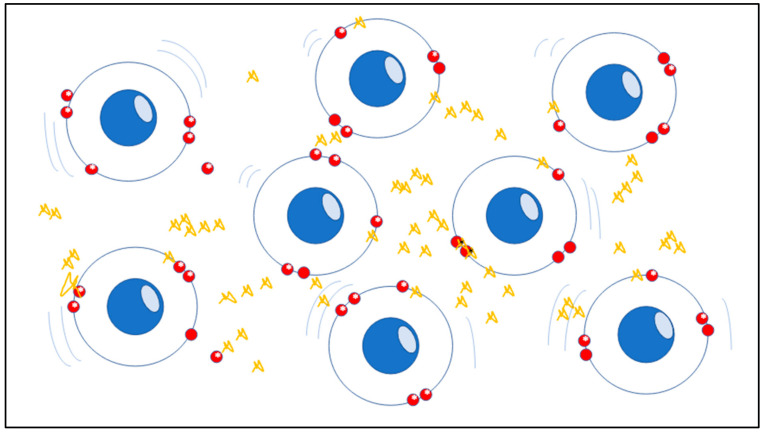
The picture represents the unstable atomic compound of mitochondria, the structure of the matter is indicated by the precise order of electrons and nuclei in atoms and molecules and is based on electron spin as a crucial moment which has physical significance in chemistry. When atoms tend to lose electrons, the red spheres, they start being negatively charged becoming unstable as ROS (Gargiulo Isacco Ciro and Nguyen CD Kieu).

**Figure 2 diagnostics-13-00813-f002:**
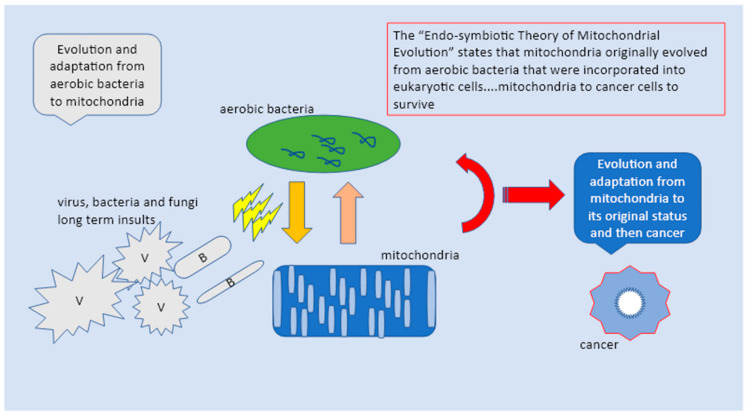
The “Endo-symbiotic Theory of Mitochondrial Evolution” states that mitochondria originally evolved from aerobic bacteria that were incorporated into eukaryotic cells. The mitochondria in response to pathogens return to their original state as independent bacteria and upon constant insults switch into cancer cells to survive the aggression.

**Figure 3 diagnostics-13-00813-f003:**
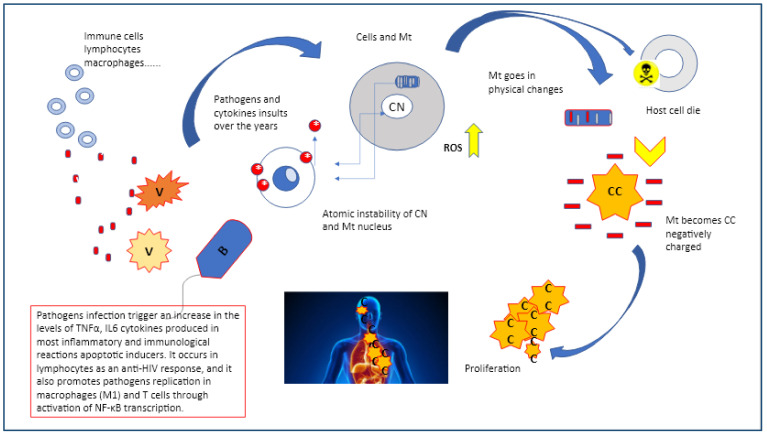
Pathogens as virus or bacteria may start a chain of events that trigger the mitochondria reverse process which takes place on multiple levels, initiating from the atomic compartment of both cell and mitochondria nuclei.

**Figure 4 diagnostics-13-00813-f004:**
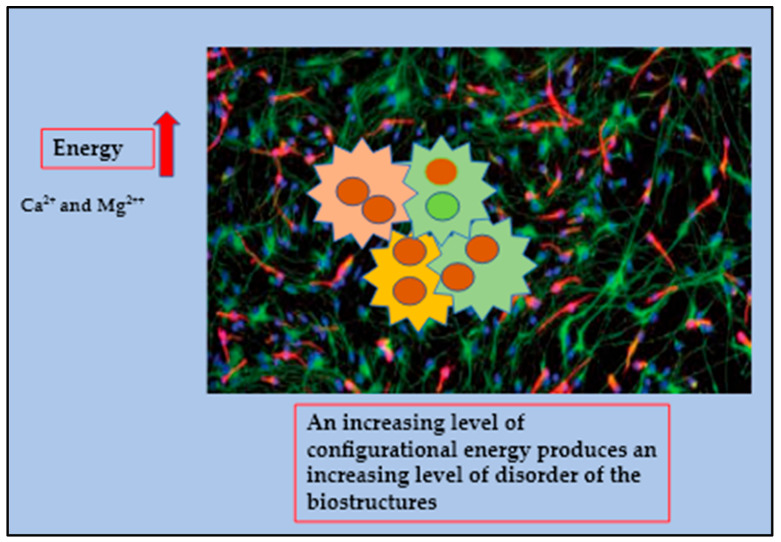
The cancer microenvironment is depicted as chaotic system characterized by an extremely high consumption of energy which in turns produces an increasing level of disorder of the biostructures which is reflected by the huge amount of Ca^2+^ and Mg^2++^ accumulated within the cancer intracellular compartment.

## Data Availability

All experimental data to support the findings of this study are available by contacting the corresponding author upon request. The authors have annotated the entire data building process and empirical techniques presented in the paper.

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
