# Peer review of "From Pathogens to Cancer: Are Cancer Cells Evolved Mitochondrial Super Cells?"

_diagnostics, 2023, doi:10.3390/diagnostics13040813_

Round 1

Reviewer 1 Report

1.The word "Mitochondria nuclei" need to be check under -The sub molecular structure of the mitochondria and cell nucleus.

2.Sentence end should be clear "(diet, environment...)"

3. Mentioning about oncogenesis the suitability of photosynthesis in sentence "In addition, redox reactions may affect multiple phases from signaling between molecules bound to DNA with potentially key effect on cell cycle up to photosynthesis" ?

4.There are many enzymes in antioxidant enzyme system which maintain the redox balance some of them are Superoxide dismutase (SOD), glutathione peroxidase (GPX) and catalase etc which can be highlighted.

5.The information under "The bacterial origin of mitochondria the common genetic traits with virus and bacteria" can be revised with more focus on antioxidant defence system as most free radicals are generated at mitochondria and cells have their own defence mechanism against the ROS.

6. The interaction of human papilloma virus (HPV), HIV, helicobacter pylori (Hp) need to be better justified with the role of mitochondria.

7. The connection of cancer cells evolved mitochondrial super cells has to be more precise and rational.

Author Response

Dear Reviewer

1. The word "Mitochondria nuclei" needs to be checked under -The sub-molecular structure of the mitochondria and cell nucleus.

Thanks to the reviewer's comments we have proceeded to change as per the suggestion.

2.Sentence end should be clear "(diet, environment...)"

Thanks to the reviewer comments we have proceeded to change as per the suggestion.

3. Mentioning oncogenesis the suitability of photosynthesis in the sentence "In addition, redox reactions may affect multiple phases from signaling between molecules bound to DNA with potentially key effect on cell cycle up to photosynthesis"?

Thanks to the reviewer's comments we have proceeded to change as per the suggestion.

“In addition, redox reactions may affect multiple phases including the signaling between molecules bound to DNA with potentially key effect on cell cycle up to the production of free energy from the dissociation of the water molecule”.

4. There are many enzymes in the antioxidant enzyme system that maintain the redox balance some of them are Superoxide dismutase (SOD), glutathione peroxidase (GPX), and catalase, etc which can be highlighted.

Thanks to the reviewer's comments we have proceeded to change as per the suggestion.

“An adjunctive aspect in this picture is the role that few enzymes such as superoxide dismutase (SOD), glutathione peroxidase (GPx) and catalase (CAT) may eventually play in the energy mechanism getting and storing available energy from oxidation processes, important in the adenosine-triphosphate (ATP) intracellular high-energy-related compartments. In addition, redox reactions may affect multiple phases including the signaling between molecules bound to DNA with potentially key effects on the cell cycle up to the production of free energy from the dissociation of the water molecule [7,8]. Intriguingly, several outcomes have pointed to the connection between persistent inflammation, oxidative stress, and carcinogenesis; those studies revealed the presence and relationship between Epstein–Barr virus/human papillomavirus (EBV/HPV) coinfection and GPx), SOD, and CAT level in several types of cancer. SOD the most important antioxidant enzyme in aerobic cells is in charge to eliminate superoxide radicals and catalyzes the dismutation of hydrogen peroxide and molecular oxygen. The GPx the key of hydrogen peroxide-scavenging converts this molecule to water, both work to maintain oxidants level within acceptable limits protecting the cells and mitochondria DNA damages” [8]

5. The information under "The bacterial origin of mitochondria the common genetic traits with virus and bacteria" can be revised with more focus on the antioxidant defense system as most free radicals are generated at mitochondria and cells have their own defense mechanism against the ROS.

Thanks to the reviewer's comments we have proceeded to change as per the suggestion and everything is highlighted in yellow.

6. The interaction of human papillomavirus (HPV), HIV, and Helicobacter pylori (Hp) needs to be better justified with the role of mitochondria.

Thanks to the reviewer's comments we have proceeded to change as per the suggestion and everything is highlighted in yellow in the main test.

7. The connection of cancer cells evolved to mitochondrial supercells has to be more precise and rational.

Thanks to the reviewer's comments we have proceeded to change as per the suggestion and everything is highlighted in yellow in the main test, section

Are cancer cells super-evolved mitochondria?

We proposed that cancer cells are newly supercells derived directly from mitochondria through a long-term mutation process. Eventually, it seems that mitochondria differentiation toward cancerous cells is marked by several molecular and genetic reprogramming steps and phases over a long period of time. By this, mitochondria with the support of either endogenous or exogenous mechanisms tend to organize a definitive structural displacement from the host cells to become completely autonomous. These achievements of structural autonomy are a heritage encoded and therefore promoted by their own genetic makeup, something that mitochondria share with bacteria and viruses. The same as bacteria and viruses, mitochondria went through indefinite numbers of evolutionary stages from free-living archeo-bacteria to becoming an integrated bio-system of the host cell. From this symbiosis both parts gained something, host cells evolved into a modern eukaryote lineage while mitochondria reached the capacity of performing higher and more complex functions, eventually, both evolved sharing integrated division mechanisms. Given the immense amount of time that viruses, bacteria, and mitochondria hosts have been coexisting it would be no surprise that the mitochondria have evolved as well and through a high metabolic reprogramming process have reached the capacity of nullifying their threats to survive at the expense of the host cell. This would support a polyphyletic origin of the mitochondrial Krebs and raises the question of what forces drove the evolution of the mitochondrial proteome [11, 31,32].

The scenario discloses what the mitochondria’s metabolic defense program indicates as “pathogens’ steady manipulation” as an effective strategy adopted by mitochondria to survive. If we consider, that the Warburg shift could benefit both the host and the pathogens whilst raising mild hypoxia as a trick used by mitochondria through evolution. In this perspective, there is an evolutionary rationale as to why mitochondria would modulate Krebs cycle intermediates to ensure its own replication, including lipid and protein metabolism, equally seen as a behavior to control pathogens' aggressiveness.

Reviewer 2 Report

The paper presents an interesting idea, where mithocondria could be responsible for cancer developing. Considering that mithocondria were ancestrally bacteria, changes in their DNA could retourn to active bacteria. In fact, redox aspects in mithocondria could be very interesting and produce diseases.

1) Line 113/114.- A reference should be given for "of the 1,000 proteins identified in mithocondria, 40% are of bacterial origen (reference).

2) Line 116.- It is not clear why a Jacobida model is used for bacterial studies. In fact, bacteria are procariote cells, whereas Jacobida are eucariotic cells. Can be better explained?

3) Line 353.- DS should be substituted by dS, or if integrated by deltaS.

Author Response

The paper presents an interesting idea, where mitochondria could be responsible for cancer development. Considering that mitochondria were ancestrally bacteria, changes in their DNA could return to active bacteria. In fact, redox aspects in mitochondria could be very interesting and produce diseases.

1) Line 113/114.- A reference should be given for "of the 1,000 proteins identified in mitochondria, 40% are of bacterial origin (reference).

We Thanks the reviewer for the suggestion reference has been added to new ref 11

2) Line 116.- It is not clear why a Jacobida model is used for bacterial studies. In fact, bacteria are procaryote cells, whereas Jacobida is eucaryotic cells. Can be better explained?

Jakobids have the relics of the mitochondrial-tRNAGly discriminator base switch. It has been used as a model to confirm the transition from an archeo-prokaryote+archeo-eukariote that met each other and eventually formed the nowadays human cells. Jakobida has a mitochondrial tRNAGly isoacceptor in a state of transition between a bacterial type with U73 and a eukaryotic type A73 containing eukaryotic-type mito-tRNAGly.

3) Line 353.- DS should be substituted by dS, or if integrated by deltaS.

We thank the reviewer for the suggestion we corrected as per the reviewer's suggestion.

Reviewer 3 Report

Balzanelli et al. hypothesize an interesting and intriguing proposition to depict cancer cells as evolved mitochondria super cells.

  In fact, this paper brings various insights to link and explore factors behind the complexity and heterogeneity of tumors.   However, important suggestions may be helpful for this manuscript for better impact.   1. A paragraph of molecular and cellular heterogeneity of tumor can be included.   2. A separate section of Warburg effects and tumor with a focus on metabolic reprogramming.   3. A review of existing data can be summarized on the cancer cells with no mitochondrial DNA and hallmarks of growth, proliferative and aggressiveness.   4. If alone mitochondria driven super cancer cells, then many organisms with mitochondria are highly resistant to cancer cells. Hence, tumor heterogeneity across species as xeno-tumor heterogeneity may be discussed in the context of mitochondria.    5. A preclinical and clinical perspective of this proposition may be highlighted. 

Author Response

reviewer 3
  In fact, this paper brings various insights to link and explore factors behind the complexity and heterogeneity of tumors.   However, important suggestions may be helpful for this manuscript for better impact.   
    1.    A paragraph on molecular and cellular heterogeneity of tumors can be included.  
We thank the reviewer for the help and the suggestions. We have added a paragraph about the topic
    2.    A separate section on Warburg effects and tumors with a focus on metabolic reprogramming.  
Cancerous cells, similarly to mitochondria and bacteria are highly adaptable to low oxygen microenvironments due to vascular vessel hemodynamics and metabolically evolved to adapt to the given oxygen environment by low coupled oxidative phosphorylation (OXPHOS) activity using anaerobic glycolysis. In addition, same as bacterial cells, tumor cell decide their proliferating directions, through genetic mutations and tumor-associated microenvironments.
    3.    A review of existing data can be summarized on the cancer cells with no mitochondrial DNA and hallmarks of growth, proliferative, and aggressiveness.  
Contrary to conventional wisdom, functional mitochondria are essential for the cancer cell. Although mutations in mitochondrial genes are common in cancer cells, they do not inactivate mitochondrial energy metabolism but rather alter the mitochondrial bioenergetic and biosynthetic state. 
    4.    If alone mitochondria drove super cancer cells, then many organisms with mitochondria are highly resistant to cancer cells. Hence, tumor heterogeneity across species as xeno-tumor heterogeneity may be discussed in the context of mitochondria.  (Lin, Y.-H.; Lim, S.-N.; Chen, C.-Y.; Chi, H.-C.; Yeh, C.-T.; Lin, W.-R. Functional Role of Mitochondrial DNA in Cancer Progression. Int. J. Mol. Sci. 2022, 23, 1659. https://doi.org/10.3390/ijms23031659).
The presence of mitochondria in several organisms may drive resistance to cancer cells.
I would answer yes, some wild animals like elephants and whales who live longer than humans are quite immune to cancer, and few scientists highlighted a possible mechanism however this would not fit with the current article being too vast and out from the topic. However, several species are known to be extremely cancer resistant. These include the naked mole rat, blind mole rat, elephant, and bowhead whale. The age of onset of cancer also varies greatly depending on the lifespan of the species. The evolutionary pressure to evolve efficient anticancer mechanisms is very strong. An animal developing cancer prior to its reproductive age would leave no progeny. Thus, animals evolved efficient mechanisms to delay the onset of tumors until post-reproductive age. Hence, cancer becomes frequent in aged animals where it is no longer subjected to natural selection. As a consequence long-lived animals are expected to have more efficient anticancer defenses to keep them cancer-free for longer. The slower metabolism of the largest mammals may lead to lower levels of cellular damage and mutations and thus contribute to lower cancer incidence. However, no data is yet available on how or whether metabolism indeed contributes to the cancer incidence in these species. It would be of great interest to understand the molecular mechanisms of cancer resistance in elephants and whales as these could potentially be translated to improve cancer resistance in humans.

A major determinant of mutation rate is DNA repair fidelity. Evidence begins to accumulate that cancer-resistant and long-lived species may have more efficient DNA repair. Long-lived species were reported to more efficiently form p53-binding protein 1 (53BP1) foci for a given amount of DNA damage suggestive of a greater capacity to detect DNA damage88. Furthermore, genome and transcriptome sequencing of long-lived animals show that multiple genes involved in DNA repair are expressed at higher levels89,90, or display the signature of positive selection. Mammalian species evolved a diverse set of anticancer mechanisms. Not all species have equal protection. Large and long-lived animals are more resistant to cancer. Some of the mechanisms that evolved are common among multiple cancer-resistant species while others only evolved in individual clades. For example, mammals with a body mass greater than 5-10 kg have all evolved repression of telomerase activity and replicative senescence. The mechanisms which have evolved in even larger species are only beginning to be understood. Elephants have evolved pseudogene duplications of the TP53 gene that may lead to an increased apoptotic response, while much larger whales do not seem to use that strategy. Mechanisms of cancer resistance found in small-bodied, long-lived animals are very diverse but all act at the early stages of cancer progression. Naked mole rats have evolved HMM-HA that restricts cell proliferation and arrests the growth of premalignant cells. Blind mole rats also express HMM-HA, but do not display ECI and instead have evolved the CCD mechanisms that trigger cell death mediated by IFN secretion in response to hyperplasia.

The reason for such diversity in tumor suppressive mechanisms is that the need for more efficient anticancer defenses has arisen independently in different phylogenetic groups. As species evolved larger body mass and/or longer lifespans, depending on their ecology, the tumor suppressor mechanisms had to adjust to become more efficient. In each case, the ecology and unique requirements of individual species would determine the outcome. The evolutionary process works with what is available; for example, a bird’s wing has evolved from an upper limb of a terrestrial animal rather than by creating a new appendage92. Similarly, in the case of two subterranean rodents, the naked mole rat and the blind mole rat, these species independently evolved HMM-HA, likely as an adaptation to subterranean lifestyle to confer stronger and more flexible skin, which constantly rubs against the walls of their burrows. Later this adaptation may have been co-opted to confer tumor resistance and longevity.

5. A preclinical and clinical perspective of this proposition may be highlighted. 
    We thank the reviewer for these suggestions, We have added information and new     paragraphs: A preclinical and clinical perspective and Current advances